# Novel CYP11A1-Derived Vitamin D and Lumisterol Biometabolites for the Management of COVID-19

**DOI:** 10.3390/nu14224779

**Published:** 2022-11-11

**Authors:** Shariq Qayyum, Radomir M. Slominski, Chander Raman, Andrzej T. Slominski

**Affiliations:** 1Department of Dermatology, University of Alabama at Birmingham, Birmingham, AL 35294, USA; 2Research Program in Men’s Health: Aging and Metabolism, The Center for Clinical Investigation, Brigham and Women’s Hospital, Harvard Medical School, 221 Longwood Avenue, Boston, MA 02115, USA; 3Department of Biological Chemistry and Molecular Pharmacology, Harvard Medical School, Boston, MA 02115, USA; 4Department of Genetics, Informatics Institute, University of Alabama at Birmingham, Birmingham, AL 35294, USA; 5Pathology and Laboratory Medicine Service, VA Medical Center, Birmingham, AL 35294, USA

**Keywords:** vitamin D, lumisterol, SARS-CoV-2, anti-inflammatory, ACE2, M^pro^, RdRp

## Abstract

Vitamin D deficiency is associated with a higher risk of SARS-CoV-2 infection and poor outcomes of the COVID-19 disease. However, a satisfactory mechanism explaining the vitamin D protective effects is missing. Based on the anti-inflammatory and anti-oxidative properties of classical and novel (CYP11A1-derived) vitamin D and lumisterol hydroxymetabolites, we have proposed that they would attenuate the self-amplifying damage in lungs and other organs through mechanisms initiated by interactions with corresponding nuclear receptors. These include the VDR mediated inhibition of NFκβ, inverse agonism on RORγ and the inhibition of ROS through activation of NRF2-dependent pathways. In addition, the non-receptor mediated actions of vitamin D and related lumisterol hydroxymetabolites would include interactions with the active sites of SARS-CoV-2 transcription machinery enzymes (M^pro^;main protease and RdRp;RNA dependent RNA polymerase). Furthermore, these metabolites could interfere with the binding of SARS-CoV-2 RBD with ACE2 by interacting with ACE2 and TMPRSS2. These interactions can cause the conformational and dynamical motion changes in TMPRSS2, which would affect TMPRSS2 to prime SARS-CoV-2 spike proteins. Therefore, novel, CYP11A1-derived, active forms of vitamin D and lumisterol can restrain COVID-19 through both nuclear receptor-dependent and independent mechanisms, which identify them as excellent candidates for antiviral drug research and for the educated use of their precursors as nutrients or supplements in the prevention and attenuation of the COVID-19 disease.

## 1. Introduction

COVID-19 is still the top health issue in the world. Vaccines approved for COVID-19 provide protection against some strains of SARS-CoV-2; however, new mutant strains develop in continuity and some of them escape immunity provided by current vaccines [1]. The infection from SARS-CoV-2 has severe adverse outcomes with a significantly higher mortality rate than influenza. The major cause of death in COVID-19 is acute respiratory distress syndrome (ARDS) caused by cytokine storm [2,3]. This enhanced hyperactivated innate immune response against the virus causes severe damage to the patient’s body/organs which might be fatal (Figure 1). In this mini-review we will discuss how vitamin D and its derivatives can be helpful against the infection caused by SARS-CoV-2.

Vitamin D, a prohormone, is a fat soluble secosteroid, which is formed in the skin after the absorption of UVB energy by the B ring of 7-dehydrocholesterol (7DHC) [4,5,6]. The prolonged exposure of 7-DHC to UVB leads to the phototransformation of pre-vitamin D3 to tachysterol and lumisterol [4,5,6]. It can also be ingested from the diet and supplements. As a prohormone, it must be activated to exert the biological activity. In the classical activation pathway vitamin D3 is metabolized by several cytochromes P450 (CYPs) enzymes before being transformed to its known active form 1,25-dihydroxyvitamin D3 (1,25(OH)_2_D3) [7,8,9,10]. 1,25(OH)_2_D3 activates the nuclear vitamin D (VDR) receptor, which controls not only body calcium metabolism [9,11,12,13] but also many important physiological functions, including the regulation of the innate and adaptive immunity [6,9,14,15,16,17]. Vitamin D can be activated by two pathways known as a canonical, with sequential hydroxylation at C25 and C1α [18], and a non-canonical, activated by CYP11A1 [19,20]. The canonical pathway includes the metabolism of vitamin D3 to 25-hydroxyvitamin D3 (25(OH)D3) by CYP2R1 and CYP27A1 in the liver and final C1α hydroxylation in the kidney to the biologically active form 1,25(OH)_2_D3 by CYP27B1 [7,8,9]. This pathway also operates in the peripheral tissues, including skin [8,18,21,22,23,24].

The phenotypic effect of 1,25(OH)_2_D3 is predominantly mediated through an interaction with the VDR leading to the transcriptional activation of more than 3000 genes [4,6,8,9,10]. Non-genomic regulatory actions for 1,25(OH)_2_D3 were also described [6,8,10]. 1,25(OH)_2_D3, in addition of regulating body calcium metabolism, also regulates diverse functions on the systemic, tissue, and cellular levels [4,6,8,9,10,11,12,13,14,15,18,19,20]. Of general interest are the anti-oxidative and anti-inflammatory properties of 1,25(OH)_2_D3, which have been appreciated for almost two decades [4,6,8,9,10,11,12,13,14,15,18,19,20]. The latter includes the downregulation of pro-inflammatory cytokines production through the inhibition of NFkB [6,12,13,14].

Recently discovered non-canonical pathways of vitamin D activation are initiated by an obligatory enzyme of steroidogenesis, CYP11A1, in a complex process involving several CYPs and producing more than a dozen of hydroxyderivatives [8,20,25,26,27]. CYP11A1 is not only expressed in adrenals, placenta, and gonads [28] but also in immune cells [29] and other peripheral organs including skin [19,30]. The CYP11A1-derived hydroxyderivatives are non-calcemic or low calcemic [31,32,33,34] and can therefore be used at high concentrations for therapeutic purposes [35,36,37,38,39]. They are also detectable in natural products including honey [40] and human serum [25,41,42]. Similarly, to 1,25(OH)_2_D3, they can alter gene expression by binding on the genomic site of the VDR [33,43,44,45,46]. CYP11A1-derived vitamin D3 hydroxyderivatives also bind to other nuclear receptors, including aryl hydrocarbon receptor (AhR) [47,48,49], retinoic acid orphan receptors (ROR)α and γ [44,50], liver X receptors (LXR)α and β [51] and can change their expression and activities [27]. The CYP11A1-derived hydroxymetabolites of vitamin D3, including 20(OH)D3 and 20,23(OH)_2_D3, have demonstrated anti-inflammatory and anti-oxidative effects [27,36,42,46,52,53,54,55,56], which are similar to the effects described for the classical active form of vitamin D3, 1,25(OH)_2_D3 [6,12,13,14,18,19,20].

Novel pathways of 7DHC transformation by CYP11A1 [30,57,58], with the further phototransformation of the 5.7-dienal products to corresponding secosteroids [27,59,60,61,62,63,64] and the hydroxylations of lumisterol by CYP11A1 and CYP27A1, were also discovered [65,66,67], with their products being biologically active and detectable in human body and acting on RORα and γ as inverse agonists and as agonists on LXR α and β and on the non-genomic site of the VDR [51,66,68]. Most recently, an enzymatic activation of tachysterol was reported with the metabolic products acting on the VDR, AhR, LXRs, and RORs [63]. These metabolites exert similar biological effects as the classical active form of vitamin D [1,25(OH)_2_D3] and they have their unique activity pattern towards various nuclear receptors, aside of the classical VDR/RXR complex [9].

Although SARS-CoV-2 infection in human cells involves multiple factors, in this review we are focusing on two main interactions listed below. The spike protein (S) of SARS-CoV-2 facilitates the entry of the virus into human cells by engaging angiotensin-converting enzyme 2 (ACE2) as their entry receptor [69] and further cellular serine protease TMPRSS2 is used for priming of S protein [70,71,72]. The association between ACE2 and Spike protein is critically important and current vaccines (mRNA) are developed to inhibit this interaction [73]. The actions of CYP11A-derived vitamin D3-hydroxymetabolites, canonical 1,25(OH)_2_D3 and lumisterol hydroxymetabolites with SARS-CoV-2 replication machinery enzymes were previously explored [74], which included molecular modeling on classical vitamin D compounds [75]. The significance of these studies is further discussed in this review.

The SARS-CoV-2 virus replicates within host cells using its cellular and enzymatic components. M^pro^, also termed 3CL protease, is a 33.8-kDa cysteine protease which helps in the maturation of functional polypeptides involved in the assembly of replication-transcription machinery [76,77,78]. M^pro^ digests the polyprotein at no less than 11 conserved sites, starting with the autolytic cleavage of this enzyme itself from pp1a and pp1ab, which are individual nonstructural proteins essential for viral genome replication [76]. Another enzyme important to the life cycle of SARS-CoV-2 is RdRp (RNA-dependent RNA polymerase), which catalyzes the replication [79] of RNA from an RNA template. SARS-CoV-2 use an RdRp complex for the replication of their genome and for the transcription of their genes [79]. M^pro^ and RdRp are enzymes required for viral replication and are not homologous to any gene in the human genome. Hence, they are very attractive targets for the development of anti-viral drugs against COVID-19. Therefore, we will further discuss how the hydroxymetabolites of vitamin D and of lumisterol can counter SARS-CoV-2 infection at different stages of this process.

## 2. CYP11A1-Derived Vitamin D and Lumisterol Hydroxymetabolites Exert Anti-Inflammatory and Antioxidant Effects

The cytokine storm is a response to viral infection, which causes immune cells to release several pro-inflammatory cytokines/chemokines (interferons, interleukins 1, 6 and 17, chemokines, colony-stimulating factors, and tumor necrosis factors (TNF)), leading to hyper inflammation and organ damage [80,81,82]. This process in the lung leads to acute lung injury and ARDS. Along with = ARDS, another factor which plays a role in damage to the tissue and cells is oxidative stress. It is secondary to the production of reactive oxygen species (ROS) and reactive nitrogen species (RNS) [83,84,85]. CYP11A1-derived vitamin D3 and lumisterol hydroxymetabolites exhibit potent anti-inflammatory activities through the inhibition of IL-1, IL-6, IL-17, TNFα and INFγ production and/or other pro-inflammatory pathways [25,36,37,43,46,50,52,54,55,56,68], which are similar to those mediated by classical 1,25(OH)_2_D3 [4,6,8,9,10,11,12,13,14,15,18,19,20]. The anti-inflammatory effects of active forms of vitamin D can be mediated through the downregulation of NF-κΒ, involving action on VDR and inverse agonism on RORγ leading to the attenuation of Th17 responses (Figure 1B). These compounds also induce anti-oxidative and reparative responses with mechanism of action involving the activation of NRF2 and p53 signaling pathways [27,42,52,53,68,86]. Interestingly, the anti-viral role of NRF2 is also recognized [87]. The use of 1,25(OH)_2_D3 has its limitations because of the toxicity that includes hypercalcemia [7,88]. However, CYP11A1-derived 20(OH)D3, 20(OH)D2, and 20,23(OH)_2_D3 are not calcemic even at very high doses [31,32,33,34]. Hence, vitamin D and its metabolites can be used as economic nutritional supplements to counter the effects of the SARS-CoV-2 infection like cytokine storm [86,89].

## 3. Inhibition of the Interaction between ACE2 and SARS-CoV-2 Spike RBD

The ACE2 interacts with the receptor-binding domain (RBD) region of the spike protein [90,91]. SARS-CoV-2 and SARS-CoV-2 RBD are typically in standing up state and resting state, respectively. SARS-COV-2 RBD has higher binding affinity to ACE2, but its lying-down state makes it less accessible to ACE2 and other inhibitory or neutralizing agents [90,91]. The use of host protease (furin, TMPRSS2, cathepsins etc.) for its activation helps as a strategy to overcome the lying down state and maintaining its high binging affinity to ACE2 [90,91,92]. This interaction is critical and important for drug development against COVID-19 infection. Molecular modeling and simulation [93] evaluated the binding of vitamin D3 and its hydroxyderivatives to SARS-CoV-2 RBD, and their potential to inhibit its interaction with ACE2. The study showed that vitamin D3 and its hydroxyderivatives can function as inhibitors of TMPRSS2 and inhibit the SARS-CoV-2 receptor binding domain (RBD) binding to the ACE2 [93]. Molecular dynamics (MD) simulations for the interactions of 1,25(OH)_2_D3 have shown the favorable binding free energy of ACE2, SARS-CoV-2 RBD, and TMPRSS2 with 1,25(OH)_2_D3 [93]. The binding free energy of ACE2, SARS-CoV-2, and TMPRSS2 with 1,25(OH)_2_D3 were −18.55 ± 4.16, −16.97 ± 1.69, and −21.04 ± 1.53 kcal/mol separately, further indicating that ACE2, SARS-CoV-2 RBD, and TMPRSS2 show favorable binding with 1,25(OH)_2_D3 [93]. The predicted interaction of 1,25(OH)_2_D3 with SARS-CoV-2 RBD and ACE2 could result in the conformation and dynamical motion changes of the binding surfaces between SARS-CoV-2 RBD and ACE2, leading to the interruption of the binding of SARS-CoV-2 RBD with ACE2 [93]. The interaction of 1,25(OH)_2_D3 with TMPRSS2 also caused the conformational and dynamical motion changes of TMPRSS2, which could affect TMPRSS2 to prime SARS-CoV-2 spike proteins [93]. These studies [93] have indicated that vitamin D3 and its biologically active hydroxymetabolites have the theoretical potential to prevent the cellular entry of SARS-CoV-2 by serving as the inhibitor of TMPRSS2 and blocking the binding of SARS-CoV-2 RBD with ACE2.

Molecular modeling was also used for the virtual screening of antiviral compounds to SARS-CoV-2 non-structural proteins [94]. The described interactions between spike protein and ACE2 can be also disrupted by other compounds, including vitamin D as described by other research groups [95,96,97]. These included interactions with vitamins, retinoids, steroids, vitamin D derivatives, and dihydrotachysterol as examples. The detailed mechanisms of action were discussed in [95,96,97]. In addition, vitamin D was identified as a potential inhibitor of COVID-19 Nsp15 endoribonuclease binding sites [98].

To confirm our predictions on novel vitamin D3 and lumisterol hydroxymetabolites, we have used an inhibitor screening kit (SARS-CoV-2 inhibitor screening kit, Acro Biosytems) (Table 1). The kit uses a colorimetric ELISA platform, which measures the binding of immobilized SARS-CoV-2 S protein and biotinylated human ACE2. Top compounds with best binding energy were theoretically predicted previously [93], selected for this assay, showing the inhibition of the interaction between ACE2 and RBD (Table 1). 20(OH)L3 showed the highest level of inhibition of the RBD-ACE2 interaction followed by 25*S*27(OH)L3 and 20(OH)D3 at concentrations of 2 × 10^−7^ M. The compounds were observed to be effective in µM concentrations, which is promising for future clinical and preclinical testing.

In addition, the treatment of HaCaT keratinocytes with these hydoxymetabolites changed the expression of ACE2 and TMPRSS2 in a metabolite-specific manner. 1,20(OH)_2_D3, 1,25(OH)_2_D3, and 24(OH)L3 suppressed the expression of ACE2, and TMPRSS2 expression was inhibited by 20(OH)D3, 1,20(OH)_2_D3, 1,25(OH)_2_D3, 20(OH)L3, and 24(OH)L3 (Appendix A). This suggests that these compounds can inhibit the bonding of ACE2 and RBD not only by directly blocking the binding but also by altering the expression of these receptors’ genes.

## 4. Inhibition of the Activity of the Replication Enzymes of SARS-CoV-2

SARS-CoV-2 replicates inside the host cell after its entry. The viral particles utilize host resources, but viral replication machinery plays crucial role in its replication. These viral specific factors do not share homologies with human proteins and are therefore targets for drug development against COVID-19 [74]. We selected two SARS-CoV-2 replication enzymes, RdRp or nsp12 and 3C-like protease (3CLpro or M^pro^), based on their recognized importance for drug development [76]. Although there are reports that vitamin D and its metabolites have potential to inhibit other viral protein, we selected these proteins as we have had experimental potential to confirm our results [74]. 3CL-Chymotrypsin, such as Protease or Main protease (M^pro^), is one of the two proteolytic enzymes that helps in cleaving the replicase polyprotein 1ab in SARS-CoV-2 at 11 specific sites, the recognition sites being Leu-Gln (Ser, Ala, Gly) in order to release 12 nsps (nsp4, nsp6-16) that are essential for viral replication as well as viral assembly [3,99]. This enzyme shares no common cleavage site with any human protease and its functional importance in the life cycle of the virus makes it an attractive target for drug development. Similarly, RNA-dependent RNA Polymerase (RdRp) is an enzyme that is responsible for the replication of RNA from an RNA template [100]. RdRp is another conserved protein of retroviruses and is also a proven target for the development of antiviral drugs [79]. We performed molecular docking on the active sites of these two enzymes and found that the hydroxyderivatives of vitamin D3 and lumisterol were binding efficiently on the active sites of these enzymes with similar affinities to known therapeutics danoprevir, lopinavir, and ritonavir serving as positive controls [74]. We further confirmed the inhibition of the enzyme activity in the presence of the compounds.

Danoprevir, lopinavir, and ritonavir were used as a standard for the comparison of predicted energies of the top 10 selected compounds [74,94]. A virtuous complementarity to the M^pro^ binding pocket was observed for top compounds, which indicates a possibility that these metabolites have ability to hinder the substrate accessibility, inhibiting the enzymatic activity in process. Significant interactions between the selected metabolites and the critically important residues of the M^pro^ substrate-binding pocket were observed (Figure 2) and predicting a block to the substrate-binding pocket of COVID-19 M^pro^ (Figure 2) [74]. The detailed analysis of the residues interaction with these metabolites was reported previously [74]. The inhibition of M^pro^ was confirmed using 3CL Protease, MBP-tagged (SARS-CoV-2) Assay (BPS Biosciences). The 25(OH)L3, 24(OH)L3, and 20S(OH)7DHC being most effective at inhibiting M^pro^ activity by 10–19% at a concentration of 2 × 10^−7^ M (Figure 3A). Similarly, selected metabolites showed interactions with critically essential residues of SARS-CoV-2 RdRp (Figure 4) [74]. These sterols and secosteroids presented a similar binding pattern to inhibitor remdesivir on RdRp active sites [74,101]. The binding prototype of the compounds showed a virtuous complementarity to the SARS-CoV-2 RdRp binding pocket predicting that they can inhibit the enzymatic activity (Figure 3B). These metabolites showed inhibitory activity ranging from 40–60% at a concentration of 10^−7^ M (Figure 3B) with 25(OH)L3 with an IC_50_ of 0.5 µM followed by 1,25(OH)_2_D3 and 20S(OH)L3, which had an IC_50_ of 1 µM. Thus, our published work [74] has demonstrated that novel 7DHC, lumisterol, and vitamin D3 hydroxymetabolites have the potential to inhibit SARS-CoV-2 infection by restricting its replication cycle. Interestingly, unbiased retrospective analyses of microarray data obtained with epithelial cells indicated the anti-viral effects of 20,23(OH)_2_D3 with similar effects for 1,25(OH)_2_D3 [86]. A potential role of hydroxylumisterols appears to be strengthened by recent findings, showing their similarity in structure to 25(OH)L3, where 25-hydroxycholesterol can act as a potent SARS-CoV-2 inhibitor [102], and that cholesterol 25-hydroxylase inhibits SARS-CoV-2 [103] and oxysterols show anti-viral activity [104].

## 5. Hypothesis

Overall, the above considerations provide strong support for the ability of D3, L3, and 7DHC hydroxymetabolites to counter the different stages of SARS-CoV-2 infection and most of them are non-calcemic. These metabolites can attenuate cytokine storm and have an ability to inhibit the viral replication enzymes. A deficiency of these hydroxymetabolites may contribute to the transition of SARS-CoV-2 patients from asymptomatic to symptomatic. Vitamin D deficiency in the body will lead to reduced levels of the vitamin D hydroxymetabolites and consequently diminished capacity to attenuate cytokine storm. Anti-inflammatory effects by vitamin D and lumisterol hydroxymetabolites were observed at 0.1 µM concentration. Similarly, 20(OH)L3, 25*S*27(OH)L3, and 20(OH)D3 were able to inhibit RBD-ACE2 interaction, which is necessary for cellular entry of the virus. For M^pro^, a significant inhibition has been observed at 0.1 µM of 25(OH)D3, which is close to its plasma concentration [8]. Of note, low pre-infection 25(OH)D3 levels are associated with a higher severity of the COVID-19 illness [105]. For RdRp, the IC_50_ for 25(OH)D3 was 1.3 µM, approximately one order of magnitude above its plasma concentration. For the hydroxylumisterols tested, plasma concentrations are unknown except for 20(OH)L3 where a value of 0.25 µM has been reported [66]. Also based on the enzymology, 25(OH)L3, which had the lowest IC_50_ (0.5 µM) for the inhibition of RdRp, is likely to have substantially higher concentrations. Interestingly, similar in structure 25(OH)cholesterol has been shown to have anti-SARS-CoV-2 activities [102,103], while cholesterol 25-hydroxylase generated anti-inflammatory environments [106] and oxysterols are recognized for anti-viral activities [104].

Therefore, defects in vitamin D or lumisterol delivery either orally as nutrients or supplements or their production in the skin after UVB exposure can lead to the deficiency of corresponding hydroxymetabolites; which have demonstrated anti-viral potential [74,86,93]. This family of compounds contains dozens of molecules [27], which can influence the different stages of SARS-CoV-2 infection. This will require further validation, as has been carried out for other molecules, including classical vitamin D3 derivatives [48,75,94,95,96,97,98]. Vaccines against SARS-CoV-2 are clearly a major advance in controlling COVID-19; however, new viral variants emphasize the need for alternative therapeutic or nutritional approaches. Therefore, consideration for novel vitamin D and L3 metabolites for anti-viral drugs that could attenuate COVID-19 is warranted.

## 6. Concluding Remarks

There are reports demonstrating a strong association for the pre-infection deficiency of vitamin D in hospitalized COVID-19 patients [100] and increased disease severity and mortality [102,103]. The oral supplementation of vitamin D may affect SARS-CoV-2 infection outcomes. Several clinical trials are ongoing, assessing the ability of vitamin D to prevent COVID-19 infection and disease severity [104,105,106,107,108,109,110], showing its importance in managing COVID-19. However, the precise mechanism of vitamin D action against COVID-19 is still unresolved. Here, we have formulated a hypothesis for the mechanism of action of vitamin D and lumisterol hydroxymetabolites against COVID-19. It includes the enzymatic activation of vitamin D3 or sterol precursors with receptor-independent [74,93] or nuclear receptor-dependent activities downstream of VDR, LXR, and AhR activation or inverse agonism on RORs [27,51,86]. Similarly, tachysterol and its metabolites represent additional candidates for at least receptor-mediated activities [63]. We also acknowledge other mechanism-oriented work on classical vitamin D hydroxyderivatives in the prevention or therapy of COVID-19 that has been reported or reviewed recently [75,96,111,112,113,114,115,116,117]. Therefore, further clinical testing for their therapeutic use would represent an important step in understanding the beneficial actions of vitamin D3, lumisterol, and possibly tachysterol derivatives [63,97] in COVID-19 with significant implications for the nutritional approach. Vitamin D, lumisterol, or tachysterol ingested through the diet or as supplements would serve as prohormones for further activation in the body towards biologically active forms.

## 7. Patent

Patent application pending (WO2022006446A1), which includes the experimental portion of this work.

## Figures and Tables

**Figure 1 nutrients-14-04779-f001:**
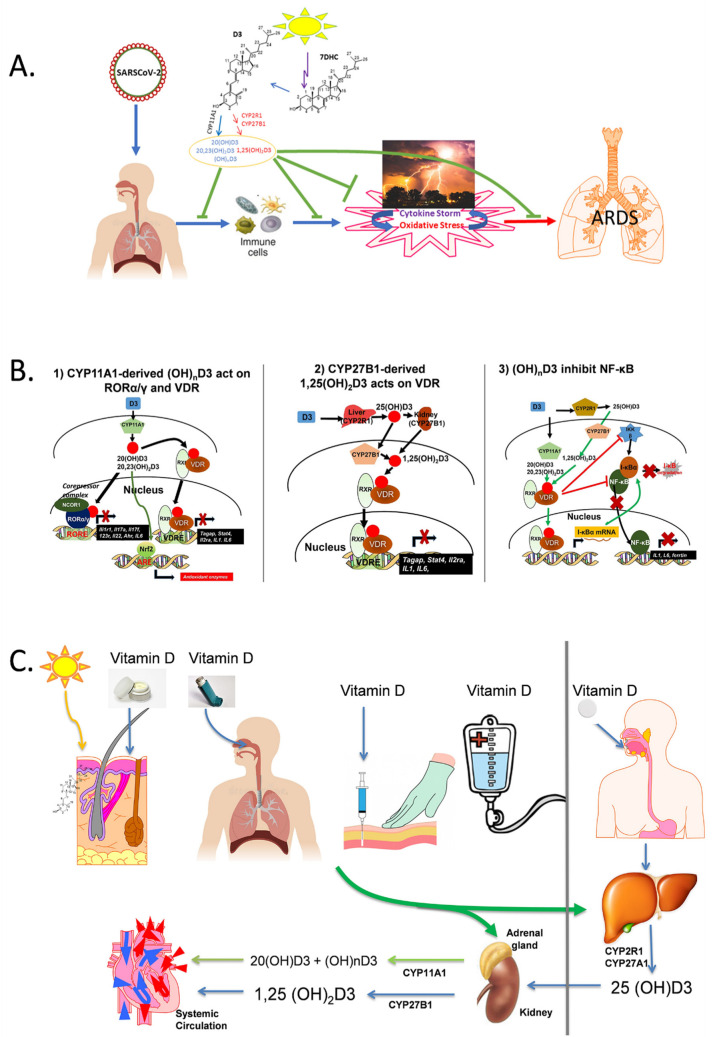
Possible mechanisms by which vitamin D can counteract the COVID-19 illness. In panel (**A**) it is proposed that the novel hydroxyderivatives of vitamin D3, in similar manner as 1,25(OH)_2_D3, inhibit cytokine storm and oxidative stress, with net attenuating effect on ARDS and multiorgan failure induced by COVID-19. Panel (**B**) proposes a mechanism of action of canonical and non-canonical vitamin D-hydroxyderivatives. Vitamin D signaling in mononuclear cells involves the activation of the VDR or inverse agonism on RORγ with downstream inhibition of inflammatory genes and the suppression of oxidative stress through the activation of NRF2. VDR, vitamin D receptor; RXR, retinoid X receptor; ROR, retinoic acid orphan receptor, RORE, ROR response element; ARE, antioxidant response element; VDRE, vitamin D response element; NRF2, transcription factor NF-E2-related factor 2. Panel (**C**) shows how different routes of vitamin D delivery impact vitamin D hydroxylation/activation patterns. Reprinted with permission from the publisher [86].

**Figure 2 nutrients-14-04779-f002:**
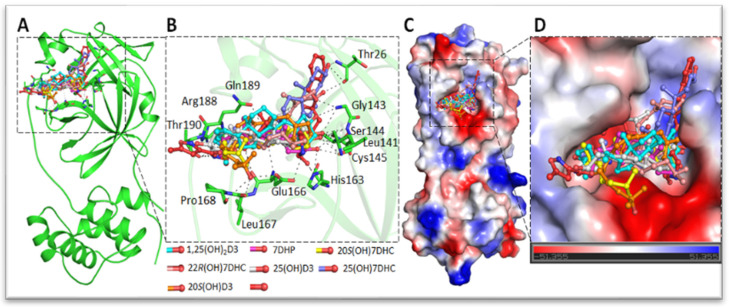
The binding pattern of identified compounds with SARS-CoV-2 M^pro^. (**A**) structural representation of the protein in complex with selected sterols and secosteroids. (**B**) selected compounds blocking the binding pocket and making significant interactions with the functionally important residues of SARS-CoV-2 M^pro^. (**C**) surface representation of conserved substrate-binding pocket of SARS-CoV-2 M^pro^ in complex with selected compounds. (**D**) zoomed view of the substrate-binding pocket of SARS-CoV-2 M^pro^ in complex with selected compounds. Reprinted with permission from the publisher [74].

**Figure 3 nutrients-14-04779-f003:**
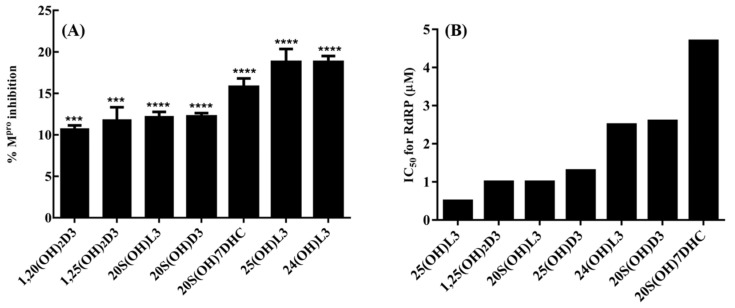
Enzyme inhibition by the selected sterols and secosteroids. (**A**) the M^pro^ enzyme inhibition by the selected metabolites at concentration of 2 × 10^−7^ M. The inhibition percentages were calculated using the formula: % inhibition = 100 × [1(X Minimum)/(Maximum–Minimum)]. Minimum = negative control without any enzyme (0% enzyme activity); Maximum = positive control with enzyme and substrate (100% enzyme activity). The test sets included enzymes, substrates, and the test compounds, and excitation at a wavelength of 360 nm and the detection of emission at a wavelength of 460 nm was observed for change in enzyme activity. The statistical significance of differences was evaluated by one-way ANOVA; *** *p*< 0.001 and **** *p* < 0.0001 for all conditions relative to ethanol blank, *n* = 3. (**B**) the RdRp enzyme activity inhibition by selected sterols and secosteroids. The inhibition percentages were calculated using the formula: % inhibition = 100 × [1 − (X-Minimum)/(Maximum–Minimum)]. Minimum = negative control without any enzyme (0% enzyme activity); Maximum = positive control with enzyme and substrate (100% enzyme activity). The statistical significance of differences was evaluated by one-way ANOVA; *** *p* < 0.001 and **** *p* < 0.0001 for all conditions relative to the ethanol blank, *n* = 3. RdRp, RNA-dependent RNA polymerase. Reprinted with permission from the publisher [74].

**Figure 4 nutrients-14-04779-f004:**
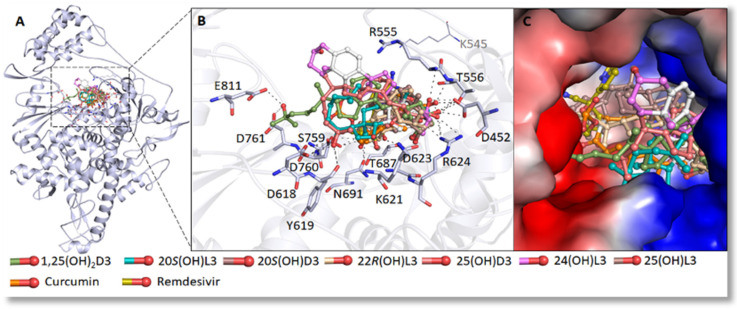
The binding pattern of identified sterols and secosteroids with SARS-CoV-2 RdRp. (**A**) structural representation of the protein in complex with selected compounds. (**B**) active site residues of the RdRp-binding pocket making significant interactions with each of the identified compounds. (**C**) surface view of the RdRp active site with the electrostatic potential from red (negative) to blue (positive) in complex with selected compounds. RdRp, RNAdependent RNA polymerase. Reprinted with permission from the publisher [74].

**Table 1 nutrients-14-04779-t001:** Inhibition of ACE2 and RBD interaction by the hydroxymetabolites. Inhibition by the selected metabolites was observed concentration of 2 × 10^−7^ M using a SARS-CoV-2 inhibitor screening kit from Acro Biosytems. The assay followed the manufacture’s protocol of the SARS-CoV-2 (B.1.617.2) Inhibitor Screening Kit (Spike RBD) (1 Innovation Way, Newark, DE 19711, USA). Data were analyzed by one way ANOVA using GraphPad Prism statistical software.

No.	Name of the Ligand	Inhibition in Enzyme Activity (%)	*p*-Value
1.	20(OH)D3	46.057	0.013
2.	1,20(OH)_2_D3	29.222	NS *
3.	1,25(OH)_2_D3	36.876	0.034
4.	20,23(OH)_2_D3	36.152	0.018
5.	24(OH)L3	32.343	0.0265
6.	20(OH)L3	74.552	0.001
7.	25*S*27(OH)L	51.722	0.005
8.	20,22(OH)_2_L3	13.074	NS *

NS * Not significant.

## Data Availability

Not applicable. The discussed papers were selected through a PubMed search and current reading of articles on COVID-19 and the role of vitamin D and sterol compounds in the therapy and prevention of the disease.

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
