# Peer review of "Novel CYP11A1-Derived Vitamin D and Lumisterol Biometabolites for the Management of COVID-19"

_nutrients, 2022, doi:10.3390/nu14224779_

Round 1

Reviewer 1 Report

The manuscript describes potential involvement and utilization of vitamin D and its related compounds to treat the COVID-19 disease.  The results the authors presented are of interest to the audience of the journal.  Regarding the data presented in Table 1, results of the known positive compounds may be more informative to the readers in addition to those of the seco-steroids.

Author Response

Reply

We thank the reviewer for his time and effort to evaluate our paper.

Table 1 is used as an example of the vitamin D derivatives activity on of ACE2 and RBD interaction. Following reviewer request we have added the missing information on positive controls (danoprevir, lopinavir and ritonavir) on lines 208-209.

Changes are marked in red and yellow

Reviewer 2 Report

Thank you for the opportunity to read this manuscript on the very relevant topic of Covid 19.

Unfortunately, the way this review has been conducted is not robust

The abstract provides no methods - how were studies chosen?

Nor is there a main discussion point

The introduction has too much vitamin D back ground

There are no methods as to how studies were selected for the review, nor a time line etc.

The main findings presented are a summary of studies - but there has been no critical analysis of presented studies, nor differentiation of the different kinds of evidence - i.e in vitro, animal, human, observational versus interventional. 

Author Response

We thank the reviewer for his time and effort to evaluate our paper.

To properly address the critique relating to abstract and discussion we have modified the title to indicate selective nature of the review and hypothesis, and the main discussion point, which is also indicated in different properly marked titles of the sections.

As relates to methods in the abstract. This is a review, and such subsection is not required for review paper.

To address critique on methods as to how studies were selected for the review, including a time line. We added explanatory sentences on lines 322-434. The main time line was since the eruption of COVID-19 epidemics up to the present time.

As relates to vitamin D we disagree, overview on new vitamin D activation pathways (discovered by the senior author) select this review from many others focused predominantly on COVID-19 or role of classical vitamin D compounds.

The current version includes critical analysis of presented studies and differentiate  different kinds of evidence.

Reviewer 3 Report

The manuscript from Qayyum and colleagues is a mini-review containing also original experimental data. The authors deal with molecular mechanisms through which vitamin D and lumisterol hydroxymetabolites exert potentially protective actions against SARS-CoV-2 infection and COVID-19. Three main mechanisms (corresponding to the three main paragraphs) are discussed: (i) the anti-inflammatory and anti-oxidant effects, (ii) the inhibition of the interaction between ACE2 and the Spike RBD, and (iii) the inhibition of the activity of replication enzymes of SARS-COV-2.

The manuscript is clearly organized and written. My main concern is that the authors wrote it focusing exclusively on their own papers. About half of the references (at least 52 out of 110 total) are self-citations. I acknowledge that the vast majorities of studies on non-canonical vitamin D metabolites are from the authors’ group, but – as written in the manuscript  – the discussed mechanisms of action also apply to “canonical” vitamin D biometabolites (e.g. 25(OH)D3 and 1,25(OH)2D3). These mechanisms have been described by many others groups, often before the authors. In a general review –  such as this one presents itself – these papers/work from other authors should be cited and discussed. More specifically:

Paragraph 2. Vitamin D and lumisterol hydroxymetabolites exert anti-inflammatory and anti-oxidant effects. To propose that the inhibition of inflammatory cytokine production, downregulation of NFkB and counteracting oxidative stress are beneficial against the hyper-inflammatory reaction and cytokine storm associated with severe COVID-19 the authors only cited their own papers. Although their group is leading in the study of CYP11A1 derived metabolites, the same pathways were shown to be activated by “canonical” vitamin D metabolites – and proposed as beneficial for fighting COVID-19 - by several other groups before. The authors can easily find previous relevant work on these topics to include in this paragraph.   

Paragraph 3. Inhibition of the interaction between ACE2 and RBD. The whole paragraph is based on authors’ ref 93, but other authors reported on the same topic.  Below a list - which may not be exhaustive – of related papers:

-          A large-scale computational screen identifies strong potential inhibitors for disrupting SARS-CoV-2 S-protein and human ACE2 interaction. Singh A, Dhar R.J Biomol Struct Dyn. 2021 May 17:1-14. doi: 10.1080/07391102.2021.1921034.

-          Molecular Simulations suggest Vitamins, Retinoids and Steroids as Ligands of the Free Fatty Acid Pocket of the SARS-CoV-2 Spike Protein. Shoemark DK, Colenso CK, Toelzer C, Gupta K, Sessions RB, Davidson AD, Berger I, Schaffitzel C, Spencer J, Mulholland AJ. Angew Chem Int Ed Engl. 2021 Mar 22;60(13):7098-7110. doi: 10.1002/anie.202015639.

-          The impact of calcitriol and estradiol on the SARS-CoV-2 biological activity: a molecular modeling approach. Mansouri A, Kowsar R, Zakariazadeh M, Hakimi H, Miyamoto A. Sci Rep. 2022 Jan 13;12(1):717. doi: 10.1038/s41598-022-04778-y

-          Discovery of adapalene and dihydrotachysterol as antiviral agents for the Omicron variant of SARS-CoV-2 through computational drug repurposing. Fidan O, Mujwar S, Kciuk M.Mol Divers. 2022 May 4:1-13. doi: 10.1007/s11030-022-10440-6

Paragraph 4. Inhibition of the activity of the replication enzymes of SARS-COV-2. Also this paragraph is based on one authors’ papers (refs 74). In that paper the authors selected two SARS-COV-2 enzymes, RdRP or nsp12 and 3C-like protease (3CLpro or Mpro) to perform molecular docking experiments. However, other authors performed molecular docking experiments selecting other viral proteins and found vitamin D metabolites as potential inhibitors. For example:

-          Vitamin D is a potential inhibitor of COVID-19: In silico molecular docking to the binding site of SARS-CoV-2 endoribonuclease Nsp15. Shalayel MH, Al-Mazaideh GM, Aladaileh SH, Al-Swailmi FK, Al-Thiabat MG. Pak J Pharm Sci. 2020 Sep;33(5):2179-2186.PMID: 33824127

Other points/suggestions:

-          The three figures included are all three already published. I would find it more useful – at least for Figure 1 -  a reworking presenting first (side by side) the canonical and non-canonical pathways of Vitamin D metabolism, then the intracellular pathways activated. It would also be helpful to describe in more detail these pathways in the text and to specify whether other vitamin D metabolites have only overlapping activities with 1,25(OH)2D3 or also unique activities (not shared with 1,25(OH)2D3). Finally, a panel/figure summarizing the three main mechanisms described in the text by which vitamin D (and derivatives) may reduce COVID-19 illness.

-          ref 75, Mok et al, is a preprint posted in 2020 but never published in a peer reviewed journal. Citations that are more appropriate could be found.

-          Lines 134-138 and 280-282 are not clear to understand, may be they can be rephrased

-          title paragraph 3 should be “… and Sars-CoV-2 Spike RDB”  

Author Response

General points

We greatly appreciate the reviewer critique, which allowed to greatly improve the manuscript. His/her time and effort are greatly appreciated.

As relates to self citations, many of the papers are co-authored by the senior author but come and are lead by other senior investigators. The field is new. To address this critique and critique on “canonical pathway” we have changed the title to add “: selective review and hypothesis”, which also point to the uniqueness of this selective analysis vs classical vitamin D derivatives.

We also added refrences requested by the reviewer. We must emphasize that we faced a problem, with formatted version that lost connection to reference manager, which required manual insertion of the refrences.

We agree many papers followed the same hypothesis as that presented in fig.1. however, these were after publication of reference 86.

Following the reviewer request we have indicated this an added several papers in the conclusion section.

Reply

We agree many papers on classical forms of vitamin D have been mechanistically analyzed for possible COVID-19 prevention and therapy.

Following the reviewer request we have indicated this an added several papers on lines: 164-166; 295-286; 313-318.

Reply to specific comments

  1. The relevant references suggested by the reviewer are included in the manuscript.
  2. The unique and overlapping effects of vitamin D metabolites are mentioned in the introduction section, we tried to be concise as it is a mini review but as per suggestion we have  expanded the explanation in the text.

  3. Appropriate references are added to manuscript according to the reviewer's suggestion
  4. Rephrased as per reviewer’s request
  5. Modified as per reviewer’s suggestion
  6.  

Round 2

Reviewer 3 Report

I thank the authors for their kind appreciation of my work. Unfortunately, they did not addressed the majority of my concerns.

My main concern was that the authors focused exclusively on their own papers and my indication was to cite and discuss evidence from other authors whenever appropriate, i.e.:

(i) when the mechanisms of action the authors describe also apply to “canonical” vitamin D biometabolites (e.g. 25(OH)D3 and 1,25(OH)2D3) and have been described by others groups, even before the authors. This was especially referred to paragraph 2, describing the anti-inflammatory and anti-oxidant effects of vitamin D and lumisterol hydroxymetabolites

The authors’ reply was the addition of the word “selective” in the title. Not a single word or reference has been added to this paragraph. The authors also argued that “ many papers followed the same hypothesis as that presented in fig.1. however, these were after publication of reference 86”.  I apologize if I was not clear enough but this is not what I meant. I wanted to point out that the discussed mechanisms – e.g.  inhibition of inflammatory cytokine production, downregulation of NFkB – have long been known to contribute to the immunomodulatory activity of canonical vitamin D metabolites. A reader new to the field, reading this paragraph, could deduce that these mechanisms of actions have been mainly established through the cited papers (all from the authors’ group or with the authors’ contribution), which is not the case. I would also like to point out that a more careful bibliographic search will show that ref 86 is not the first that associated these mechanisms with a possible beneficial effect in COVID-19. I still believe that, even in a selective review, the work on CYP11A1 derived metabolites should be framed in the context of previous (or concomitant) literature data on canonical metabolites.

(ii) when discussing the direct interaction of vitamin D metabolites with viral proteins, proposed by other groups in addition to the authors (paragraph 3 and 4)

The authors simply cited the references I suggested, without spending a single word on the work of colleagues. One of the refs is also misquoted: ref 98 refers to the theme treated in paragraph 4, but was cited in paragraph 3. More attention and an effort to describe and explain to the reader relevant work from other groups will substantially improve the manuscript.

Other points:

No reworking of the figure has been performed.

No details have been added in the text to describe the different pathways activated by the different metabolites, except for the generic sentence “Hence, these metabolites show parallel effects to classical active form of vitamin D [1,25(OH)2D3] as well as they have their unique activity pattern towards various nuclear receptors”. Unless a specific space limit is present, more details would be useful and should go along with the revised figure. In other words: a new “ad hoc” figure (as detailed in my previous comments) adequately introduced/described in the text, instead of the already published Figure 1, would contribute to improve the quality of the manuscript.

Lines 134-138 and 280-282 are now lanes 137-139 and 291 – 293 respectively, but have not been rephrased.   

Author Response

Reply 1

We appreciate the reviewer effort to improve our presentation. We agree with the reviewer that the anti-oxidative and anti-inflammatory properties  have been appreciated for decades. These are described in many reviews.

We now discuss these properties  in the text with referral to reviews on the subject. These sections are separated from those describing activities of CYP11A1 derived compounds and appropriate recognition is given for these mechanisms of action, including downregulation of NFkB. Thus, the work on CYP11A1 derived metabolites has been framed in the context of previous literature data on canonical metabolites as requested.

To further emphasize our focus on CYP11A1 derived metabolites, we indicate this already in the title. In fact, this review was invited to talk about these new metabolites, since for the classical ones, have been discussed in numerous excellent reviews.

For additional clarification, publication 86 is focused on novel CYP11A1 derived compounds, which although show phenotypic similarity to 1,25D3 but the receptor mediated mechanism of action is different. For example, for anti-inflammatory activity, they act as inverse agonists of RORγ. Also, as it relates to other receptors such as AhR, 1,25D3 binds with much lower affinity than 20,23(OH)2D3, with the latter and 17,20,23D3 acting as best ligands from the secosteroidal family. However, this property is outside of this paper focused on COVID-19

Reply 2

We have corrected misquotation and extended the discussion on these interactions. However, we hope that the reviewer agrees with us, that this is focused on CYP11A1-derived compounds, and the classical compounds have already been described thoroughly by others.

Reply 3

We believe that the figure 1 is comprehensive and proper permission from the publisher was already obtained. Any change may require additional permission without guaranteeing that such would be obtained. It is focused on CYP11A1 derived D3-deriovatives. However, we provide more detailed description of the concept.

Lines 137-139 and 291-293, have been corrected. We apologize for the omission

Final comments

We hope that the reviewer will accept our changes and find the manuscript satisfactory, as two other reviewers did, after first round of corrections.

We also want to mention, that  during move of the text to the Nutrients form, references are being disconnected from the end-notes, which makes extremely difficult to make any major changes in references in the text, except on the end of the manuscript.

The changes are tracked in the manuscript using tools.
